# Scalable Global Optimization via Local Bayesian Optimization

**David Eriksson**
Uber AI
eriksson@uber.com

**Michael Pearce**
University of Warwick
m.a.l.pearce@warwick.ac.uk

**Jacob R Gardner**
Uber AI
jake.gardner@uber.com

**Ryan Turner**
Uber AI
ryan.turner@uber.com

**Matthias Poloczek**
Uber AI
poloczek@uber.com

## Abstract

Bayesian optimization has recently emerged as a popular method for the sample-efficient optimization of expensive black-box functions. However, the application to high-dimensional problems with several thousand observations remains challenging, and on difficult problems Bayesian optimization is often not competitive with other paradigms. In this paper we take the view that this is due to the implicit homogeneity of the global probabilistic models and an overemphasized exploration that results from global acquisition. This motivates the design of a *local* probabilistic approach for global optimization of large-scale high-dimensional problems. We propose the `TuRBO` algorithm that fits a collection of local models and performs a principled global allocation of samples across these models via an implicit bandit approach. A comprehensive evaluation demonstrates that `TuRBO` outperforms state-of-the-art methods from machine learning and operations research on problems spanning reinforcement learning, robotics, and the natural sciences.

## 1 Introduction

The global optimization of high-dimensional black-box functions—where closed form expressions and derivatives are unavailable—is a ubiquitous task arising in hyperparameter tuning [36]; in reinforcement learning, when searching for an optimal parametrized policy [7]; in simulation, when calibrating a simulator to real world data; and in chemical engineering and materials discovery, when selecting candidates for high-throughput screening [18]. While Bayesian optimization (BO) has emerged as a highly competitive tool for problems with a small number of tunable parameters (e.g., see [13, 35]), it often scales poorly to high dimensions and large sample budgets. Several methods have been proposed for high-dimensional problems with small budgets of a few hundred samples (see the literature review below). However, these methods make strong assumptions about the objective function such as low-dimensional subspace structure. The recent algorithms of Wang et al. [45] and Hernández-Lobato et al. [18] are explicitly designed for a large sample budget and do not make these assumptions. However, they do not compare favorably with state-of-the-art methods from stochastic optimization like `CMA-ES` [17] in practice.

The optimization of high-dimensional problems is hard for several reasons. First, the search space grows exponentially with the dimension, and while local optima may become more plentiful, global optima become more difficult to find. Second, the function is often heterogeneous, making the task of fitting a global surrogate model challenging. For example, in reinforcement learning problems with sparse rewards, we expect the objective function to be nearly constant in large parts of the search space. For the latter, note that the commonly used global Gaussian process (GP) models [13, 46]

implicitly suppose that characteristic lengthscales and signal variances of the function are constant in the search space. Previous work on non-stationary kernels does not make this assumption, but these approaches are too computationally expensive to be applicable in our large-scale setting [37, 40, 3]. Finally, the fact that search spaces grow considerably faster than sampling budgets due to the curse of dimensionality implies the inherent presence of regions with large posterior uncertainty. For common myopic acquisition functions, this results in an overemphasized exploration and a failure to exploit promising areas.

To overcome these challenges, we adopt a *local* strategy for BO. We introduce trust region BO (TuRBO), a technique for global optimization, that uses a collection of simultaneous *local* optimization runs using independent probabilistic models. Each local surrogate model enjoys the typical benefits of Bayesian modeling —robustness to noisy observations and rigorous uncertainty estimates— however, these local surrogates allow for heterogeneous modeling of the objective function and do not suffer from over-exploration. To optimize globally, we leverage an implicit multi-armed bandit strategy at each iteration to allocate samples between these local areas and thus decide which local optimization runs to continue.

We provide a comprehensive experimental evaluation demonstrating that TuRBO outperforms the state-of-the-art from BO, evolutionary methods, simulation optimization, and stochastic optimization on a variety of benchmarks that span from reinforcement learning to robotics and natural sciences. An implementation of TuRBO is available at `https://github.com/uber-research/TuRBO`.

## 1.1    Related work

BO has recently become the premier technique for global optimization of expensive functions, with applications in hyperparameter tuning, aerospace design, chemical engineering, and materials discovery; see [13, 35] for an overview. However, most of BO's successes have been on low-dimensional problems and small sample budgets. This is not for a lack of trying; there have been many attempts to scale BO to more dimensions and observations. A common approach is to replace the GP model: Hutter et al. [19] uses random forests, whereas Snoek et al. [38] applies Bayesian linear regression on features from neural networks. This neural network approach was refined by Springenberg et al. [39] whose BOHAMIANN algorithm uses a modified Hamiltonian Monte Carlo method, which is more robust and scalable than standard Bayesian neural networks. Hernández-Lobato et al. [18] combines Bayesian neural networks with Thompson sampling (TS), which easily scales to large batch sizes. We will return to this acquisition function later.

There is a considerable body of work in high-dimensional BO [8, 21, 5, 44, 14, 45, 32, 26, 27, 6]. Many methods exist that exploit potential additive structure in the objective function [21, 14, 45]. These methods typically rely on training a large number of GPs (corresponding to different additive structures) and therefore do not scale to large evaluation budgets. Other methods exist that rely on a mapping between the high-dimensional space and an unknown low-dimensional subspace to scale to large numbers of observations [44, 27, 15]. The BOCK algorithm of Oh et al. [29] uses a cylindrical transformation of the search space to achieve scalability to high dimensions. *Ensemble Bayesian optimization* (EBO) [45] uses an ensemble of additive GPs together with a batch acquisition function to scale BO to tens of thousands of observations and high-dimensional spaces. Recently, Nayebi et al. [27] have proposed the general HeSBO framework that extends GP-based BO algorithms to high-dimensional problems using a novel subspace embedding that overcomes the limitations of the Gaussian projections used in [44, 5, 6]. From this area of research, we compare to BOCK, BOHAMIANN, EBO, and HeSBO.

To acquire large numbers of observations, large-scale BO usually selects points in batches to be evaluated in parallel. While several batch acquisition functions have recently been proposed [9, 34, 43, 47, 48, 24, 16], these approaches do not scale to large batch sizes in practice. TS [41] is particularly lightweight and easy to implement as a batch acquisition function as the computational cost scales linearly with the batch size. Although originally developed for bandit problems [33], it has recently shown its value in BO [18, 4, 22]. In practice, TS is usually implemented by drawing a realization of the unknown objective function from the surrogate model's posterior on a discretized search space. Then, TS finds the optimum of the realization and evaluates the objective function at that location. This technique is easily extended to batches by drawing multiple realizations as (see the supplementary material for details).

Evolutionary algorithms are a popular approach for optimizing black-box functions when thousands of evaluations are available, see Jin et al. [20] for an overview in stochastic settings. We compare to the successful covariance matrix adaptation evolution strategy (CMA-ES) of Hansen [17]. CMA-ES performs a stochastic search and maintains a multivariate normal sampling distribution over the search space. The evolutionary techniques of recombination and mutation correspond to adaptions of the mean and covariance matrix of that distribution.

High-dimensional problems with large sample budgets have also been studied extensively in operations research and simulation optimization, see [11] for a survey. Here the successful trust region (TR) methods are based on a local surrogate model in a region (often a sphere) around the best solution. The trust region is expanded or shrunk depending on the improvement in obtained solutions; see Yuan [49] for an overview. We compare to BOBYQA [31], a state-of-the-art TR method that uses a quadratic approximation of the objective function. We also include the Nelder-Mead (NM) algorithm [28]. For a $d$-dimensional space, NM creates a $(d+1)$-dimensional simplex that adaptively moves along the surface by projecting the vertex of the worst function value through the center of the simplex spanned by the remaining vertices. Finally, we also consider the popular quasi-Newton method BFGS [50], where gradients are obtained using finite differences. For other work that uses local surrogate models, see e.g., [23, 42, 1, 2, 25].

## 2 The trust region Bayesian optimization algorithm

In this section, we propose an algorithm for optimizing high-dimensional black-box functions. In particular, suppose that we wish to solve:

$$\text{Find } \mathbf{x}^* \in \Omega \text{ such that } f(\mathbf{x}^*) \leq f(\mathbf{x}), \ \ \forall \mathbf{x} \in \Omega,$$

where $f : \Omega \to \mathbb{R}$ and $\Omega = [0,1]^d$. We observe potentially noisy values $y(\mathbf{x}) = f(\mathbf{x}) + \varepsilon$, where $\varepsilon \sim \mathcal{N}(0, \sigma^2)$. BO relies on the ability to construct a global model that is *eventually* accurate enough to uncover a global optimizer. As discussed previously, this is challenging due to the curse of dimensionality and the heterogeneity of the function. To address these challenges, we propose to abandon global surrogate modeling, and achieve global optimization by maintaining several *independent local models*, each involved in a separate local optimization run. To achieve global optimization in this framework, we maintain multiple local models simultaneously and allocate samples via an implicit multi-armed bandit approach. This yields an efficient acquisition strategy that directs samples towards promising local optimization runs. We begin by detailing a single local optimization run, and then discuss how multiple runs are managed.

**Local modeling.** To achieve principled local optimization in the gradient-free setting, we draw inspiration from a class of TR methods from stochastic optimization [49]. These methods make suggestions using a (simple) surrogate model inside a TR. The region is often a sphere or a polytope centered at the best solution, within which the surrogate model is believed to accurately model the function. For example, the popular COBYLA [30] method approximates the objective function using a local linear model. Intuitively, while linear and quadratic surrogates are likely to be inadequate models globally, they can be accurate in a sufficiently small TR. However, there are two challenges with traditional TR methods. First, deterministic examples such as COBYLA are notorious for handling noisy observations poorly. Second, simple surrogate models might require overly small trust regions to provide accurate modeling behavior. Therefore, we will use GP surrogate models within a TR. This allows us to inherit the robustness to noise and rigorous reasoning about uncertainty that global BO enjoys.

**Trust regions.** We choose our TR to be a hyperrectangle centered at the best solution found so far, denoted by $\mathbf{x}^\star$. In the noise-free case, we set $\mathbf{x}^\star$ to the location of the best observation so far. In the presence of noise, we use the observation with the smallest posterior mean under the surrogate model. At the beginning of a given local optimization run, we initialize the *base side length* of the TR to $L \leftarrow L_{\text{init}}$. The actual side length for each dimension is obtained from this base side length by rescaling according to its lengthscale $\lambda_i$ in the GP model while maintaining a total volume of $L^d$. That is, $L_i = \lambda_i L / (\prod_{j=1}^d \lambda_j)^{1/d}$. To perform a single local optimization run, we utilize an acquisition function at each iteration $t$ to select a batch of $q$ candidates $\{\mathbf{x}_1^{(t)}, \ldots, \mathbf{x}_q^{(t)}\}$, restricted to be within the TR. If $L$ was large enough for the TR to contain the whole space, this would be

equivalent to running standard global BO. Therefore, the evolution of $L$ is critical. On the one hand, a TR should be sufficiently large to contain good solutions. On the other hand, it should be small enough to ensure that the local model is accurate within the TR. The typical behavior is to expand a TR when the optimizer "makes progress", i.e., it finds better solutions in that region, and shrink it when the optimizer appears stuck. Therefore, following, e.g., Nelder and Mead [28], we will shrink a TR after too many consecutive "failures", and expand it after many consecutive "successes". We define a "success" as a candidate that improves upon $\mathbf{x}^\star$, and a "failure" as a candidate that does not. After $\tau_{\text{succ}}$ consecutive successes, we double the size of the TR, i.e., $L \leftarrow \min\{L_{\text{max}}, 2L\}$. After $\tau_{\text{fail}}$ consecutive failures, we halve the size of the TR: $L \leftarrow L/2$. We reset the success and failure counters to zero after we change the size of the TR. Whenever $L$ falls below a given minimum threshold $L_{\text{min}}$, we discard the respective TR and initialize a new one with side length $L_{\text{init}}$. Additionally, we do not let the side length expand to be larger than a maximum threshold $L_{\text{max}}$. Note that $\tau_{\text{succ}}$, $\tau_{\text{fail}}$, $L_{\text{min}}$, $L_{\text{max}}$, and $L_{\text{init}}$ are hyperparameters of TuRBO; see the supplementary material for the values used in the experimental evaluation.

**Trust region Bayesian optimization.** So far, we have detailed a single *local* BO strategy using a TR method. Intuitively, we could make this algorithm (more) global by random restarts. However, from a probabilistic perspective, this is likely to utilize our evaluation budget inefficiently. Just as we reason about which candidates are most promising within a local optimization run, we can reason about which local optimization run is "most promising."

Therefore, TuRBO maintains $m$ trust regions *simultaneously*. Each trust region $\text{TR}_\ell$ with $\ell \in \{1, \ldots, m\}$ is a hyperrectangle of base side length $L_\ell \leq L_{\text{max}}$, and utilizes an independent local GP model. This gives rise to a classical exploitation-exploration trade-off that we model by a multi-armed bandit that treats each TR as a lever. Note that this provides an advantage over traditional TR algorithms in that TuRBO puts a stronger emphasis on promising regions.

In each iteration, we need to select a batch of $q$ candidates drawn from the union of all trust regions, and update all local optimization problems for which candidates were drawn. To solve this problem, we find that TS provides a principled solution to both the problem of selecting candidates within a single TR, and selecting candidates across the set of trust regions simultaneously. To select the $i$-th candidate from across the trust regions, we draw a realization of the posterior function from the local GP within each TR: $f_\ell^{(i)} \sim \mathcal{GP}_\ell^{(t)}(\mu_\ell(\mathbf{x}), k_\ell(\mathbf{x}, \mathbf{x}'))$, where $\mathcal{GP}_\ell^{(t)}$ is the GP posterior for $\text{TR}_\ell$ at iteration $t$. We then select the $i$-th candidate such that it minimizes the function value across all $m$ samples *and* all trust regions:

$$\mathbf{x}_i^{(t)} \in \operatorname*{argmin}_{\ell} \operatorname*{argmin}_{\mathbf{x} \in \text{TR}_\ell} f_\ell^{(i)} \text{ where } f_\ell^{(i)} \sim \mathcal{GP}_\ell^{(t)}(\mu_\ell(\mathbf{x}), k_\ell(\mathbf{x}, \mathbf{x}')).$$

That is, we select as point with the smallest function value after concatenating a Thompson sample from each TR for $i = 1, \ldots, q$. We refer to the supplementary material for additional details.

## 3 Numerical experiments

In this section, we evaluate TuRBO on a wide range of problems: a 14D robot pushing problem, a 60D rover trajectory planning problem, a 12D cosmological constant estimation problem, a 12D lunar landing reinforcement learning problem, and a 200D synthetic problem. All problems are multimodal and challenging for many global optimization algorithms. We consider a variety of batch sizes and evaluation budgets to fully examine the performance and robustness of TuRBO. The values of $\tau_{\text{succ}}$, $\tau_{\text{fail}}$, $L_{\text{min}}$, $L_{\text{max}}$, and $L_{\text{init}}$ are given in the supplementary material.

We compare TuRBO to a comprehensive selection of state-of-the-art baselines: BFGS, BOCK, BOHAMIANN, CMA-ES, BOBYQA, EBO, GP-TS, HeSBO-TS, Nelder-Mead (NM), and random search (RS). Here, GP-TS refers to TS with a global GP model using the Matérn-$5/2$ kernel. HeSBO-TS combines GP-TS with a subspace embedding and thus effectively optimizes in a low-dimensional space; this target dimension is set by the user. Therefore, a small sample budget may suffice, which allows to run $p$ invocations in parallel, following [44]. This may improve the performance, since each embedding may "fail" with some probability [27], i.e., it does not contain the active subspace even if it exists. Note that HeSBO-TS-$p$ recommends a point of optimal posterior mean among the $p$ GP-models; we use that point for the evaluation. The standard acquisition criterion EI used in BOCK and BOHAMIANN is replaced by (batch) TS, i.e., all methods use the same criterion which allows for a

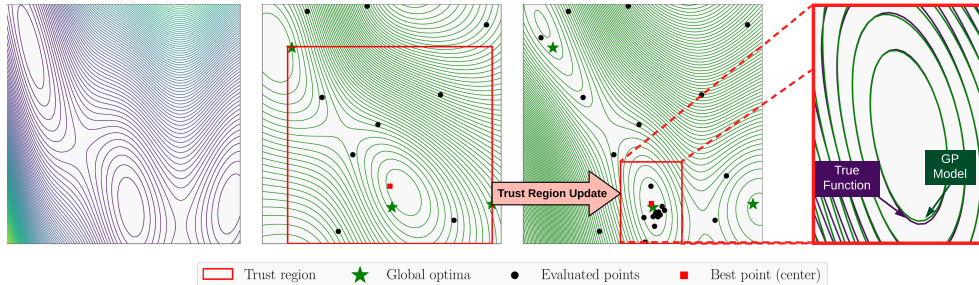

| | Trust region | ★ Global optima | ● Evaluated points | ■ Best point (center) |

Figure 1: Illustration of the `TuRBO` algorithm. **(Left)** The true contours of the Branin function. **(Middle left)** The contours of the GP model fitted to the observations depicted by black dots. The current TR is shown as a red square. The global optima are indicated by the green stars. **(Middle right)** During the execution of the algorithm, the TR has moved towards the global optimum and has reduced in size. The area around the optimum has been sampled more densely in effect. **(Right)** The local GP model almost exactly fits the underlying function in the TR, despite having a poor global fit.

direct comparison. Methods that attempt to learn an additive decomposition lack scalability and are thus omitted. `BFGS` approximates the gradient via finite differences and thus requires $d+1$ evaluations for each step. Furthermore, `NM`, `BFGS`, and `BOBYQA` are inherently sequential and therefore have an edge by leveraging all gathered observations. However, they are considerably more time consuming on a per-wall-time evaluation basis since we are working with large batches.

We supplement the optimization test problems with three additional experiments: i) one that shows that `TuRBO` achieves a linear speed-up from large batch sizes, ii) a comparison of local GPs and global GPs on a control problem, and iii) an analytical experiment demonstrating the locality of `TuRBO`. Performance plots show the mean performances with one standard error. Overall, we observe that `TuRBO` consistently finds excellent solutions, outperforming the other methods on most problems. Experimental results for a small budget experiment on four synthetic functions are shown in the supplement, where we also provide details on the experimental setup and runtimes for all algorithms.

### 3.1 Robot pushing

The robot pushing problem is a noisy 14D control problem considered in Wang et al. [45]. We run each method for a total of 10K evaluations and batch size of $q = 50$. `TuRBO`-1 and all other methods are initialized with 100 points except for `TuRBO`-20 where we use 50 initial points for each trust region. This is to avoid having `TuRBO`-20 consume its full evaluation budget on the initial points. We use `HeSBO-TS`-5 with target dimension 8. `TuRBO`-$m$ denotes the variant of `TuRBO` that maintains $m$ local models in parallel. Fig. 2 shows the results: `TuRBO`-1 and `TuRBO`-20 outperform the alternatives. `TuRBO`-20 starts slower since it is initialized with 1K points, but eventually outperforms `TuRBO`-1. `CMA-ES` and `BOBYQA` outperform the other BO methods. Note that Wang et al. [45] reported a median value of 8.3 for `EBO` after 30K evaluations, while `TuRBO`-1 achieves a mean and median reward of around 9.4 after only 2K samples.

### 3.2 Rover trajectory planning

Here the goal is to optimize the locations of 30 points in the 2D-plane that determine the trajectory of a rover [45]. Every algorithm is run for 200 steps with a batch size of $q = 100$, thus collecting a total of 20K evaluations. We use 200 initial points for all methods except for `TuRBO`-20, where we use 100 initial points for each region. Fig. 2 summarizes the performance. We observe that `TuRBO`-1 and `TuRBO`-20 outperform all other algorithms after a few thousand evaluations. `TuRBO`-20 once again starts slowly because of the initial 2K random evaluations. Wang et al. [45] reported a mean value of 1.5 for `EBO` after 35K evaluations, while `TuRBO`-1 achieves a mean and median reward of about 2 after only 1K evaluations. We use a target dimension of 10 for `HeSBO-TS`-15 in this experiment.

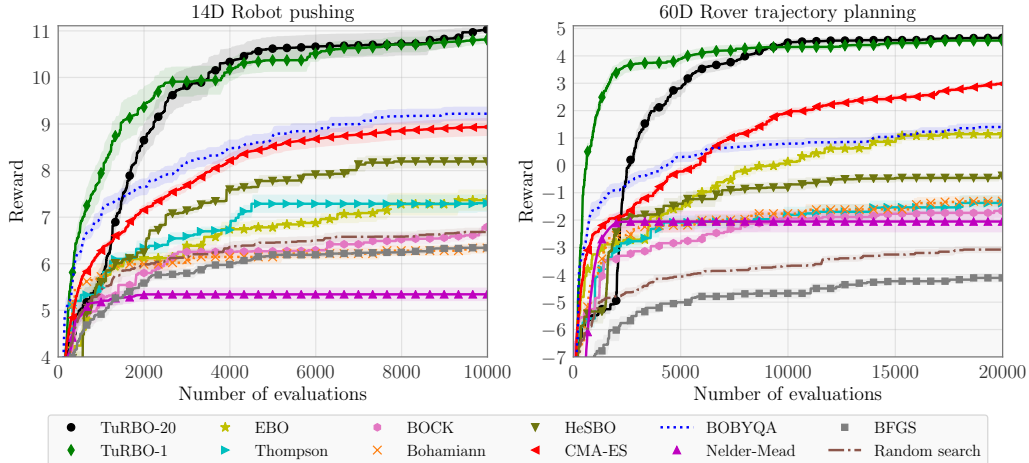

Figure 2: **14D Robot pushing (left):** `TuRBO`-1 and `TuRBO`-20 perform well after a few thousand evaluations. **60D Rover trajectory planning (right):** `TuRBO`-1 and `TuRBO`-20 achieve close to optimal objective values after 10K evaluations. In both experiments `CMA-ES` and `BOBYQA` are the runners up, and `HeSBO-TS` and `EBO` perform best among the other BO methods.

### 3.3 Cosmological constant learning

In the "cosmological constants" problem, the task is to calibrate a physics simulator[1] to observed data. The tunable parameters include various physical constants like the density of certain types of matter and Hubble's constant. In this paper, we use a more challenging version of the problem in [21] by tuning 12 parameters rather than 9, and by using substantially larger parameter bounds. We used 2K evaluations, a batch size of $q = 50$, and 50 initial points. `TuRBO`-5 uses 20 initial points for each local model and `HeSBO-TS`-4 uses a target dimension of 8. Fig. 3 (left) shows the results, with `TuRBO`-5 performing the best, followed by `BOBYQA` and `TuRBO`-1. `TuRBO`-1 sometimes converges to a bad local optimum, which deteriorates the mean performance and demonstrates the importance of allocating samples across multiple trust regions.

### 3.4 Lunar landing reinforcement learning

Here the goal is to learn a controller for a lunar lander implemented in the OpenAI gym[2]. The state space for the lunar lander is the position, angle, time derivatives, and whether or not either leg is in contact with the ground. There are four possible action for each frame, each corresponding to firing a booster engine left, right, up, or doing nothing. The objective is to maximize the average final reward over a fixed constant set of 50 randomly generated terrains, initial positions, and velocities. We observed that the simulation can be sensitive to even tiny perturbations. Fig. 3 shows the results for a total of 1500 function evaluations, batch size $q = 50$, and 50 initial points for all algorithms except for `TuRBO`-5 which uses 20 initial points for each local region. For this problem, we use `HeSBO-TS`-3 in an 8-dimensional subspace. `TuRBO`-5 and `TuRBO`-1 learn the best controllers; and in particular achieves better rewards than the handcrafted controller provided by OpenAI whose performance is depicted by the blue horizontal line.

### 3.5 The 200-dimensional Ackley function

We examine performances on the 200-dimensional Ackley function in the domain $[-5, 10]^{200}$. We only consider `TuRBO`-1 because of the large number of dimensions where there may not be a benefit from using multiple TRs. EBO is excluded from the plot since its computation time exceeded 30 days per replication. `HeSBO-TS`-5 uses a target dimension of 20. Fig. 4 shows the results for a total of 10K function evaluations, batch size $q = 100$, and 200 initial points for all algorithms.

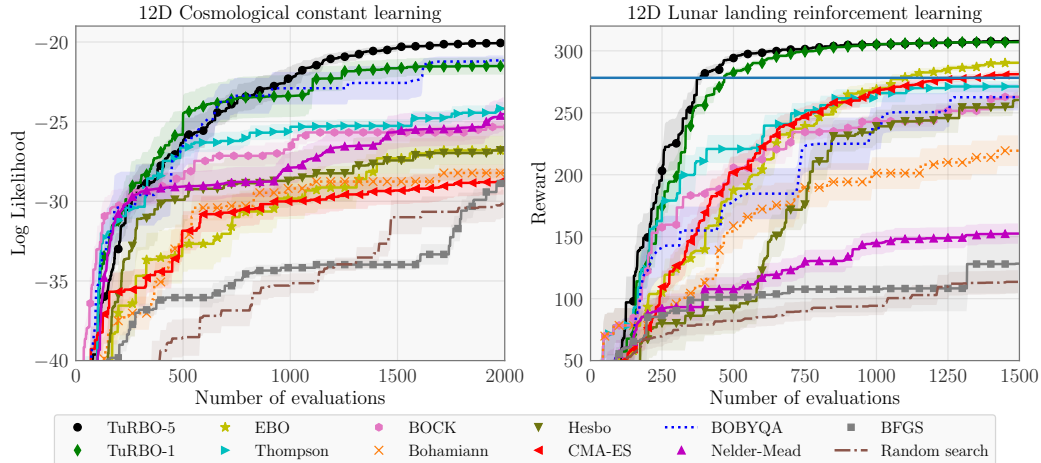

Figure 3: **12D Cosmological constant (left):** `TuRBO`-5 provides an improvement over `BOBYQA` and `TuRBO`-1. BO methods are distanced, with `TS` performing best among them. **12D Lunar lander (right):** `TuRBO`-5, `TuRBO`-1, `EBO`, and `CMA-ES` learn better controllers than the original `OpenAI` controller (solid blue horizontal line).

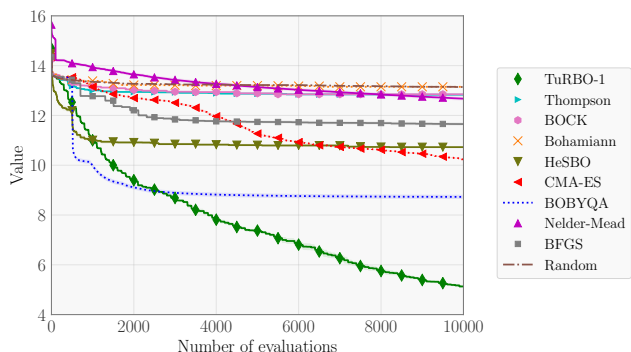

Figure 4: **200D Ackley function:** `TuRBO`-1 clearly outperforms the other baselines. `BOBYQA` makes good initial progress but consistently converges to sub-optimal local minima.

`HeSBO-TS`-5, with a target dimension of 20, and `BOBYQA` perform well initially, but are eventually outperformed by `TuRBO`-1 that achieves the best solutions. The good performance of `HeSBO-TS` is particularly interesting, since this benchmark has no redundant dimensions and thus should be challenging for that embedding-based approach. This confirms similar findings in [27]. BO methods that use a global GP model over-emphasize exploration and make little progress.

## 3.6 The advantage of local models over global models

We investigate the performance of local and global GP models on the 14D robot pushing problem from Sect. 3.1. We replicate the conditions from the optimization experiments as closely as possible for a regression experiment, including for example parameter bounds. We choose 20 uniformly distributed hypercubes of (base) side length 0.4, each containing 200 uniformly distributed training points. We train a global GP on all 4000 samples, as well as a separate local GP for each hypercube. For the sake of illustration, we used an isotropic kernel for these experiments. The local GPs have the advantage of being able to learn different hyperparameters in each region while the global GP has the advantage of having access to all of the data. Fig. 5 shows the predictive performance (in log loss) on held-out data. We also show the distribution of fitted hyperparameters for both the local and global GPs. We see that the hyperparameters (especially the signal variance) vary substantially across regions. Furthermore, the local GPs perform better than the global GP in every repeated trial. The global model has an average log loss of 1.284 while the local model has an average log loss of 1.174

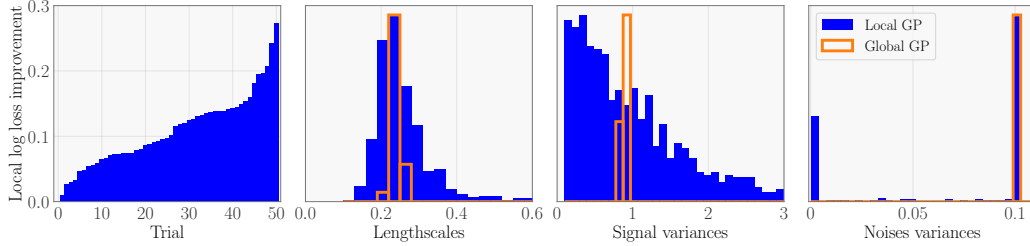

Figure 5: **Local and global GPs on log loss (left):** We show the improvement in test set log loss (nats/test point) of the local model over the global model by repeated trial. The local GP increases in performance in every trial. Trials are sorted in order of performance gain. This shows a substantial mean improvement of $0.110$ nats. **Learned hypers (right three figures):** A histogram plot of the hyperparameters learned by the local (blue) and global (orange) GPs pooled across all repeated trials. The local GPs show a much wider range of hyperparameters that can specialize per region.

across 50 trials; the improvement is significant under a $t$-test at $p < 10^{-4}$. This experiment confirms that we improve the predictive power of the models and also reduce the computational overhead of the GP by using the local approach. The learned local noise variance in Fig. 5 is bimodal, confirming the heteroscedasticity in the objective across regions. The global GP is required to learn the high noise value to avoid a penalty for outliers.

## 3.7 Why high-dimensional spaces are challenging

In this section, we illustrate why the restarting and banditing strategy of `TuRBO` is so effective. Each TR restart finds distant solutions of varying quality, which highlights the multimodal nature of the problem. This gives `TuRBO`-$m$ a distinct advantage.

We ran `TuRBO`-1 (with a single trust region) for 50 restarts on the 60D rover trajectory planning problem from Sect. 3.2 and logged the volume of the TR and its center after each iteration. Fig. 6 shows the volume of the TR, the arclength of the TR center's trajectory, the final objective value, and the distance each final solution has to its nearest neighbor. The left two plots confirm that, within a trust region, the optimization is indeed highly local. The volume of any given trust region decreases rapidly and is only a small fraction of the total search space. From the two plots on the right, we see that the solutions found by `TuRBO` are far apart with varying quality, demonstrating the value of performing multiple local search runs in parallel.

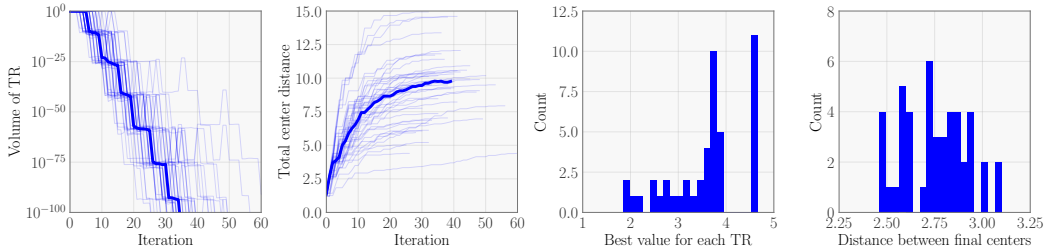

Figure 6: Performance statistics for 50 restarts of `TuRBO`-1 on the 60D rover trajectory planning problem. The domain is scaled to $[0,1]^{60}$. **Trust region volume (left):** We see that the volume of the TR decreases with the iterations. Each TR is shown by a light blue line, and their average in solid blue. **Total center distance (middle left):** The cumulative Euclidean distance that each TR center has moved (trajectory arc length). This confirms the balance between initial exploration and final exploitation. **Best value found (middle right):** The best function value found during each run of `TuRBO`-1. The solutions vary in quality, which explains why our bandit approach works well. **Distance between final TR centers (right):** Minimum distances between final TR centers, which shows that each restart leads to a different part of the space.

## 3.8 The efficiency of large batches

Recall that combining multiple samples into single batches provides substantial speed-ups in terms of wall-clock time but poses the risk of inefficiencies since sequential sampling has the advantage of leveraging more information. In this section, we investigate whether large batches are efficient for TuRBO. Note that Hernández-Lobato et al. [18] and Kandasamy et al. [22] have shown that the TS acquisition function is efficient for batch acquisition with a single global surrogate model. We study TuRBO-1 on the robot pushing problem from Sect. 3.1 with batch sizes $q \in \{1, 2, 4, \ldots, 64\}$. The algorithm takes $\max\{200q, 6400\}$ samples for each batch size and we average the results over 30 replications. Fig. 7 (left) shows the reward for each batch size with respect to the number of batches: we see that larger batch sizes obtain better results for the same number of iterations. Fig. 7 (right) shows the performance as a function of evaluations. We see that the speed-up is essentially linear.

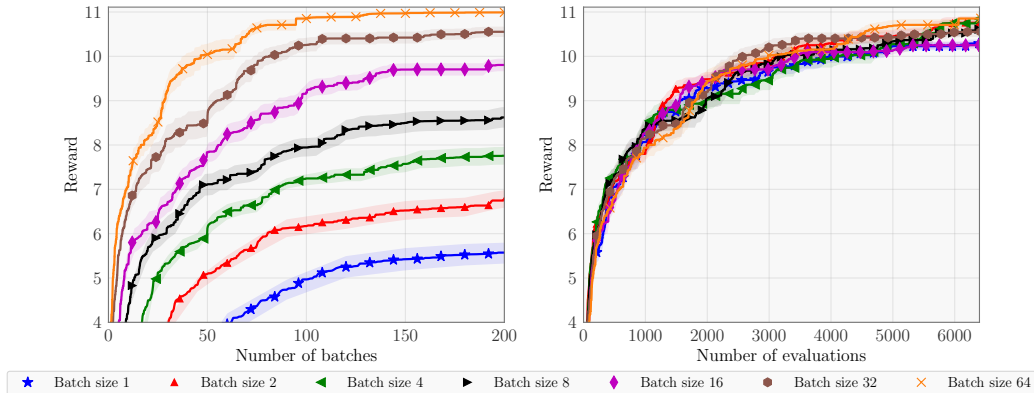

Figure 7: We evaluate TuRBO for different batch sizes. On the left, we see that larger batches provide better solutions at the same number of steps. On the right, we see that this reduction in wall-clock time does not come at the expense of efficacy, with large batches providing nearly linear speed up.

## 4 Conclusions

The global optimization of computationally expensive black-box functions in high-dimensional spaces is an important and timely topic [13, 27]. We proposed the TuRBO algorithm which takes a novel local approach to global optimization. Instead of fitting a global surrogate model and trading off exploration and exploitation on the whole search space, TuRBO maintains a collection of local probabilistic models. These models provide local search trajectories that are able to quickly discover excellent objective values. This local approach is complemented with a global bandit strategy that allocates samples across these trust regions, implicitly trading off exploration and exploitation. A comprehensive experimental evaluation demonstrates that TuRBO outperforms the state-of-the-art Bayesian optimization and operations research methods on a variety of real-world complex tasks.

In the future, we plan on extending TuRBO to learn local low-dimensional structure to improve the accuracy of the local Gaussian process model. This extension is particularly interesting in high-dimensional optimization when derivative information is available [10, 12, 48]. This situation often arises in engineering, where objectives are often modeled by PDEs solved by adjoint methods, and in machine learning where gradients are available via automated differentiation. Ultimately, it is our hope that this work spurs interest in the merits of Bayesian *local* optimization, particularly in the high-dimensional setting.

## Footnotes

[1] https://lambda.gsfc.nasa.gov/toolbox/lrgdr/

[2] https://gym.openai.com/envs/LunarLander-v2

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
