[Supplementary Material · neurips_2019_supp.pdf]

# Scalable Global Optimization via Local Bayesian Optimization Supplementary Material

**David Eriksson**
Uber AI
eriksson@uber.com

**Michael Pearce**
University of Warwick
m.a.l.pearce@warwick.ac.uk

**Jacob R Gardner**
Uber AI
jake.gardner@uber.com

**Ryan Turner**
Uber AI
ryan.turner@uber.com

**Matthias Poloczek**
Uber AI
poloczek@uber.com

In Sect. 1 we provide additional benchmarking results on synthetic problems. We explain the algorithms considered in this paper in more detail in Sect. 2. Then we describe how we leverage scalable GP regression in Sect. 3. We summarize the hyperparameters of TuRBO in Sect. 4 and give additional details on how we shrink and expand the trust regions. Thompson sampling is summarized in Sect. 5. Finally, we describe the test problems in Sect. 6 and provide runtimes for all benchmark problems in Sect. 7.

## 1 Synthetic experiments

We present results on four popular synthetic problems: Ackley with domain $[-5, 10]^{10}$, Levy with domain $[-5, 10]^{10}$, Rastrigin with domain $[-3, 4]^{10}$, and the 6D Hartmann function with domain $[0, 1]^6$. The optimizers are given a budget of $50$ batches of size $q = 10$ which results in a total of $n = 500$ function evaluations. All methods use 20 initial points from a Latin hypercube design (LHD) [8] except for TuRBO-5, where we use 10 initial points in each local region. To compute confidence intervals on the results, we use 30 runs. For HeSBO-TS we used target dimension 4 for Hartmann6 and 6 for the other benchmarks.

Fig. 1 summarizes the results. We observed a good performance for TuRBO-1 and TuRBO-5 on all test problems. TuRBO-1 and TuRBO-5 outperform other methods on Ackley and consistently find solutions close to the global optimum. The results for Levy also show that TuRBO-5 clearly performs best. However, TuRBO-1 found solutions close to the global optimum in some trials but struggled in others, which shows that a good starting position is important. On Rastrigin, TuRBO-5 performs the best. BOBYQA and BFGS perform comparably to TuRBO-1. In contrast, the 6D Hartmann function is much easier and most methods converge quickly.

Interestingly, the embedding-based HeSBO-TS algorithm performs well on Levy and Rastrigin. On the other hand, BOHAMIANN struggles compared to other BO methods, suggesting that its model fit is inaccurate compared to GP-based methods. We also observe that CMA-ES finds good solutions eventually for Ackley, Rastrigin, and Hartmann, albeit considerably slower than TuRBO. For Levy CMA-ES seems stuck with suboptimal solutions.

## 2 Algorithms background

In this section, we provide additional background on the three categories of competing optimization methods: traditional local optimizers, evolutionary algorithms, and other recent works in large-scale BO. Namely, we compare TuRBO to Nelder-Mead (NM), BOBYQA, BFGS, EBO, Bayesian optimization

Figure 1: `TuRBO` and `TuRBO-5` perform well on all synthetic benchmark problems. `HeSBO-TS` performs well on Levy and Rastrigin. `BOBYQA` and `BFGS` are competitive on Rastrigin and Hartmann6, showing that local optimization can outperform global optimization on multimodal functions.

with cylindrical kernels (`BOCK`), `HeSBO-TS`, `BOHAMIANN`, Thompson sampling with a global GP (`GP-TS`), `CMA-ES`, and random search (`RS`). This is an extensive set of state-of-the-art optimization algorithms from both local and global optimization.

For local optimization, we use the popular `NM`, `BOBYQA`, and `BFGS` methods with multiple restarts. They are all initialized from the best of a few initial points. We use the `Scipy` [6] implementations of `NM` and `BFGS` and the `nlopt` [5] implementation of `BOBYQA`.

Evolutionary algorithms often perform well for black-box optimization with a large number of function evaluations. These methods are appropriate for large batch sizes since they evaluate a population in parallel. We compare to `CMA-ES` [3] as it outperforms differential evolution, genetic algorithms, and particle swarms in most of our experiments. We use the `pycma`[1] implementation with the default settings and a population size equal to the batch size. The population is initialized from the best of a few initial points.

To the best of our knowledge, `EBO` is the only BO algorithm that has been applied to problems with large batch sizes and tens of thousands of evaluations. We also compare to `GP-TS`, `BOCK`, `HeSBO-TS`, and `BOHAMIANN`, all using Thompson sampling as the acquisition function. The original implementations of `BOCK` and `BOHAMIANN` often take hours to suggest a single point and do not support batch suggestions. This necessitated changes to use them for our high-dimensional setting with large batch sizes. To generate a discretized candidate set, we generate a set of scrambled Sobolev sequences with 5000 points for each batch.

## 3 Gaussian process regression

We further provide details on both the computational scaling and modeling setup for the GP. To address computational issues, we use GPyTorch [2] for scalable GP regression. GPyTorch follows Dong et al. [1] to solve linear systems using the conjugate gradient (CG) method and approximates the log-determinant via the Lanczos process. Without GPyTorch, running BO with a GP model for more than a few thousand evaluations would be infeasible as classical approaches to GP regression scale cubically in the number of data points.

On the modeling side, the GP is parameterized using a Matérn-$5/2$ kernel with ARD and a constant mean function for all experiments. The GP hyperparameters are fitted before proposing a new batch by optimizing the log-marginal likelihood. The domain is rescaled to $[0, 1]^d$ and the function values are standardized before fitting the GP. We use a Matérn-$5/2$ kernel with ARD for TuRBO and use the following bounds for the hyperparameters: (lengthscale) $\lambda_i \in [0.005, 2.0]$, (signal variance) $s^2 \in [0.05, 20.0]$, (noise variance) $\sigma^2 \in [0.0005, 0.1]$.

## 4 TuRBO details

In all experiments, we use the following hyperparameters for TuRBO-1: $\tau_{\text{succ}} = 3$, $\tau_{\text{fail}} = \lceil d/q \rceil$, $L_{\min} = 2^{-7}$, $L_{\max} = 1.6$, and $L_{\text{init}} = 0.8$, where $d$ is the number of dimensions and $q$ is the batch size. Note that this assumes the domain has been scaled to the unit hypercube $[0, 1]^d$. When using TuRBO-1, we consider an improvement from at least one evaluation in the batch a *success* [9]. In this case, we increment the success counter and reset the failure counter to zero. If no point in the batch improves the current best solution we set the success counter to zero and increment the failure counter.

When using TuRBO with more than one TR, we use the same tolerances as in the sequential case ($q = 1$) as the number of evaluations allocated by each TR may differ in each batch. We use separate success and failure counters for each TR. We consider a batch a success for $TR_\ell$ if $q_\ell > 0$ points are selected from this TR and at least one is better than the best solution in this TR. The counters for this TR are updated just as for TuRBO-1 in this case. If all $q_\ell > 0$ evaluations are worse than the current best solution we consider this a failure and set the success counter to zero and add $q_\ell$ to the failure counter. The failure counter is set to $\tau_{\text{fail}}$ if we increment past this tolerance, which will trigger a halving of its side length.

For each TR, we initialize $L \leftarrow L_{\text{init}}$ and terminate the TR when $L < L_{\min}$. Each TR in TuRBO uses a candidate set of size $\min\{100d, 5000\}$ on which we generate each Thompson sample. We create each candidate set by first generating a scrambled Sobolev sequence within the intersection of the TR and the domain $[0, 1]^d$. A new candidate set is generated for each batch. In order to not perturb all coordinates at once, we use the value in the Sobolev sequence with probability $\min\{1, 20/d\}$ for a given candidate and dimension, and the value of the center otherwise. A similar strategy is used by Regis and Shoemaker [10] where perturbing only a few dimensions at a time showed to substantially improve the performance for high-dimensional functions.

## 5 Thompson sampling

In this section, we provide details and pseudo-code that makes the background on Thompson sampling (TS) with GPs precise. Conceptually, TS [12] for BO works by drawing a function $f$ from the surrogate model (GP) posterior. It then makes a suggestion by reporting the optimum of the function $f$. This process is repeated independently for multiple suggestions ($q > 1$). The exploration-exploitation trade off is naturally handled by the stochasticity in sampling.

Furthermore, parallel batching is naturally handled by the marginalization coherence of GPs. Many acquisition functions handle batching by *imputing* function evaluations for the other suggested (but unobserved) points via sampling from the posterior. Independent TS for parallel batches is exactly equivalent to conditioning on imputed values for unobserved suggestions. This means TS also trivially handles *asynchronous* batch sampling [4, 7].

Note that we cannot sample an entire function $f$ from the GP posterior in practice. We therefore work in a discretized setting by first drawing a finite *candidate set*; this puts us in the same setting as the

traditional multi-arm bandit literature. To do so, we sample the GP marginal on the candidate set, and then apply regular Thompson sampling.

# 6 Test problems

In this section we provide some brief additional details for the test problems. We refer the reader to the original papers for more details.

## 6.1 Robot pushing

The robot pushing problem was first considered in Wang et al. [13]. The goal is to tune a controller for two robot hands to push two objects to given target locations. The robot controller has $d = 14$ parameters that specify the location and rotation of the hands, pushing speed, moving direction, and pushing time. The reward function is $f(\mathbf{x}) = \sum_{i=1}^{2} \|\mathbf{x}_{gi} - \mathbf{x}_{si}\| - \|\mathbf{x}_{gi} - \mathbf{x}_{fi}\|$, where $\mathbf{x}_{si}$ are the initial positions of the objects, $\mathbf{x}_{fi}$ are the final positions of the objects, and $\mathbf{x}_{gi}$ are the goal locations.

## 6.2 Rover trajectory planning

This problem was also considered in Wang et al. [13]. The goal is to optimize the trajectory of a rover over rough terrain, where the trajectory is determined by fitting a B-spline to 30 points in a 2D plane. The reward function is $f(\mathbf{x}) = c(\mathbf{x}) - 10(\|\mathbf{x}_{1,2} - \mathbf{x}_s\|_1 + \|\mathbf{x}_{59,60} - \mathbf{x}_g\|_1) + 5$, where $c(\mathbf{x})$ penalizes any collision with an object along the trajectory by $-20$. Here, $\mathbf{x}_s$ and $\mathbf{x}_g$ are the desired start and end positions of the trajectory. The cost function hence adds a penalty when the start and end positions of the trajectory are far from the desired locations.

## 6.3 Cosmological constant learning

The cosmological constant experiment uses luminous red galaxy data from the Sloan Digital Sky Survey [11]. The objective function is a likelihood estimate of a simulation based astrophysics model of the observed data. The parameters include various physical constants, such as Hubble's constant, the densities of baryonic and other forms of matter. We use the nine parameters tuned in previous papers, plus three additional parameters chosen from the many available to the simulator.

## 6.4 Lunar lander reinforcement learning

The lunar lander problem is taken from the OpenAI gym[2]. The objective is to learn a controller for a lunar lander that minimizes fuel consumption and distance to a landing target, while also preventing crashes. At any time, the state of the lunar lander is its angle and position, and their respective time derivatives. This 8-dimensional state vector $\mathbf{s}$ is passed to a handcrafted parameterized controller that determines which of 4 actions $a$ to take. Each corresponds to firing a booster engine: $a \in \{\text{nothing, left, right, down}\}$. The handcrafted control policy has $d = 12$ parameters that parameterize linear score functions of the state vector and also the thresholds that determine which action to prioritize. The objective is the average final reward over a fixed constant set of 50 randomly generated terrains, initial positions, and initial velocities. Simulation runs were capped at 1000 time steps, after which failure to land was scored as a crash.

# 7 Runtimes

In Table 1, we provide the algorithmic runtime for the numerical experiments. This is the total runtime for one optimization run, excluding the time spent evaluating the objective function. We see that the local optimizers and the evolutionary methods run with little to no overhead on all problems. The BO methods with a global GP model become computationally expensive when the number of evaluations increases and we leverage scalable GPs on an NVIDIA RTX 2080 TI. TuRBO does not only outperform the other BO methods, but runs in minutes on all test problems and is in fact more than $2000\times$ faster than the slowest BO method.

|            | Synthetic | Lunar landing | Cosmological constant | Robot pushing | Rover trajectory | Ackley-200 |
|------------|-----------|---------------|-----------------------|---------------|------------------|------------|
| Evaluations $n$ | 500 | 1500 | 2000 | 10,000 | 20,000 | 10,000 |
| Dimensions $d$ | 6 or 10 | 12 | 12 | 14 | 60 | 200 |
| TuRBO | <1 min | <1 min | <1 min | 8 min | 22 min | 10 min |
| EBO | 4 min | 23 min | 1 h | 11 d | >30 d | NA |
| GP-TS | 3 min | 6 min | 11 min | 1 h | 3 h | 1 h |
| BOCK | 6 min | 10 min | 19 min | 2 h | 7 h | 2 h |
| BOHAMIANN | 2 h | 5 h | 7 h | 20 h | 2 d | 25 h |
| NM | <1 min | <1 min | <1 min | <1 min | <1 min | <1 min |
| CMA-ES | <1 min | <1 min | <1 min | <1 min | <1 min | <1 min |
| BOBYQA | <1 min | <1 min | <1 min | <1 min | <1 min | <1 min |
| BFGS | <1 min | <1 min | <1 min | <1 min | <1 min | <1 min |
| RS | <1 min | <1 min | <1 min | <1 min | <1 min | <1 min |

Table 1: Algorithmic overhead for one optimization run for each test problem. The times are rounded to minutes, hours, or days.

## Footnotes

[1]`https://github.com/CMA-ES/pycma`

[2]gym.openai.com/envs/LunarLander-v2/