[Reviews · NeurIPS 2019]

Reviewer 1



Major * I found this paper to be very exciting, presenting a promising methodology addressing some of the most critical bottlenecks of Bayesian Optimization, with a focus on large data sets (being therefore relevant for high-dimensional BO as well, where sample sizes typically need to be substantially increased with the dimension). * With respect to the dimension indeed, several questions arose with respect to the considered hypercubes: * Page 4: Each trust region TR_ell with ell in {1,...,m} is a hypercube of side length L_ell \leq 1, and utilizes an independent local GP model. So, one is far from filling the space, right? * Page 4, about Lmin=(1/2)^6: could some more explanations be provided on the underlying rationale? * Page 4, equation just before the start of Section 3: why randomizing when the whole distribution is known and tractable? * Page 5, about "We replaced the standard acquisition criterion EI used in BOCK and BOHAMIANN by TS to achieve the required parallelism": but there exist several ways of making EI batch-sequential (see for instance a few papers dealing with this: Marmin et al. (2015): https://link.springer.com/chapter/10.1007%2F978-3-319-27926-8_4 González et al, (2016) http://proceedings.mlr.press/v51/gonzalez16a.pdf Wang et al. (2019): https://arxiv.org/pdf/1602.05149.pdf Not using these for some good reason is one thing, but putting it the way it is put here sounds like it is not possible to go batch-sequential with EI... * In the main contributions presented throughout Section 3, two main ideas are confounded here: splitting the data so as to obtain local models AND using TS as infill criterion. Which is (most) responsible for improved performances over the state of the art? Minor (selected points) * Page 1: What does "outputscales" mean? * Page 2, about "For commonly used myopic acquisition functions, this results in an overemphasized exploration and a failure to exploit promising areas.": better explaining why and/or referring to other works where this is analyzed in more detail would be nice. * Page 5, syntax issue in "Note that BFGS requires gradient approximations via finite differences, which is a fair comparison when the number of function evaluations is counted accordingly. * Throughout Section 3: is the (log) loss introduced? ******** Update afer rebuttal ********** I am happy with the way the authors addressed reviewer comments in their rebuttal, and while several points raised by the reviewing team give food for thoughts towards follow-up contributions, I feel that this paper deserves to be published in NeurIPS 2019. I do not increase my score as it is already high.

Reviewer 2



I found this paper quite interesting and I think the contribution is quite original and appealing to the community. The paper is nicely written, easy to follow and it is evaluated in a fair number of challenging scenarios and multiple methods. Mi main criticism is the lack of comparison of previous "local" Bayesian optimization methods. Bayesian optimization with a dual (local and global) strategy, or with a locally-biased strategy has been previously explored in the past by several authors. Just to give some examples: -K. P. Wabersich and M. Toussaint: Advancing Bayesian Optimization: The Mixed-Global-Local (MGL) Kernel and Length-Scale Cool Down. NIPS Workshop on Bayesian Optimization, Preprint at arxiv.org/abs/1612.03117, 2016. -Martinez-Cantin R. Funneled Bayesian optimization for design, tuning and control of autonomous systems. IEEE transactions on cybernetics. 2018 Feb 27(99):1-2. -Acerbi, L. and Ma, W. J. (2017). Practical Bayesian Optimization for Model Fitting with Bayesian Adaptive Direct Search. Proc. Advances in Neural Information Processing Systems 30 (NeurIPS ’17), Long Beach, USA. -Akrour R, Sorokin D, Peters J, Neumann G. Local Bayesian optimization of motor skills. InProceedings of the 34th International Conference on Machine Learning-Volume 70 2017 Aug 6 (pp. 41-50). JMLR. org. -McLeod, M., Roberts, S. and Osborne, M.A.. (2018). Optimization, fast and slow: optimally switching between local and Bayesian optimization. Proceedings of the 35th International Conference on Machine Learning, in PMLR 80:3443-3452 The most related to this papers are the works of Wabersich and Toussaint; and Martinez-Cantin which also splits the model and the resources in a local and global region. Wabersich even computes the local region in terms of the accuracy of the quadratic approximation, similar to a trust region for quadratic algorithms, such as BOBYQA. Also, in the introduction, it is mentioned that "... note that the commonly used global Gaussian process models implicitly suppose that characteristic lengthscales and outputscales are the function of constants in the search space...". That is also not the case in some previous works on nonstationarity in Bayesian optimization. For example, there are nonstationary kernels like the previous work from Martinez-Cantin; warping spaces like the work of Snoek et at.; or treed GPs like the works of Taddy et al. and Assael et al. - Snoek J, Swersky K, Zemel R, Adams R. Input warping for bayesian optimization of non-stationary functions. InInternational Conference on Machine Learning 2014 Jan 27 (pp. 1674-1682). - Taddy MA, Lee HK, Gray GA, Griffin JD. Bayesian guided pattern search for robust local optimization. Technometrics. 2009 Nov 1;51(4):389-401. - Assael JA, Wang Z, Shahriari B, de Freitas N. Heteroscedastic treed bayesian optimisation. arXiv preprint arXiv:1410.7172. 2014 Oct 27. In fact, the whole discussion about RL having sparse rewards and therefore requiring a nonstationay process is also the motivation of Martinez-Cantin's paper. Instead, the authors focus on the parallelization of the algorithm, which seems secondary, distract from the main point of the paper and leads to some questionable decisions in the experimental process. For example, while I praise the choice of "non-standard" algorithms for comparison, the fact that they replace the EI acquisition in BOHAMIAN and BOCK for TS, which is known to be less effective in the sequential case. Furthermore, most of the experiments presented are actually sequential, such as the robot pushing (one would probably have a single robot) or the rover planing (those plans are computed in limited onboard computers). Following with the experimental section, there are a couple of minor comments: -The results of EBO for the robot experiments are quite different from the results on the original paper, specially the variance of the rover planning. -Given that the objective of BO is sample efficiency, it would be interesting to see the results of a standard GP+EI, maybe limiting the results to few hundreds of evaluation. In theory, the 60D problem should be intractable, but the 12-14D problems could be solved with a standard GP. -Why using COBYLA instead of BOBYQA from the same author? BOBYQA should be faster as it uses quadratic functions. It assumes that the function is twice differentiable, but so does the Matern 5/2 kernel. -In all the problems. TuRBO gets a different number of initial samples. For example, in the rover case, all methods gets 200 while TuRBO-30 gets 3000, which is one order of magnitude more. This seems unfair. Regarding the method, the only comment I have is with respect to the bandit equation, which is purely based on the function sample and not the information/uncertainty on that region. That might result in a lack of exploration and poor global convergence, specially because the sample is based only on the local GP. Wouldn't be better to express the bandit equation in a exploration/exploitation dependent function such as a global acquisition function? Maybe that could explain why TuRBO requires such a large initial set. Despite all my comments, the results are quite impressive. ---- Update: Most of my concerns have been properly addressed by the authors.

Reviewer 3



Summary: This paper proposes a new Bayesian optimization strategy called TuRBO, which aims to perform global optimization via a set of local Bayesian optimization routines. The goal of TuRBO is to show good performance in both high dimensions, and with large numbers of queries/observations. This strategy uses trust region methods to adaptively constrain the domain, which using a multi-armed bandit strategy to choose between different local optimizers. A Thompson sampling approach is used to select a subsequent point given the multiple trust regions. Comments: > My main criticism is that the empirical results don’t seem to go up to very high dimensions. In the empirical results, three of the four tasks are from 10-20 dimensions, while one task is in 60 dimensions. In some of the high dimensional BO papers listed in the related work, tasks are shown from 50-120 dimensions. > It would be great to explicitly define or clarify what is meant by a “heterogeneous function”. There is a brief description involving reinforcement learning problems (in Section 1). However, I feel that this does not provide a clear description or definition of what the authors mean. > Minor: the abstract has the line “the application to high-dimensional problems with several thousand observations remains challenging”. At first pass, the phrasing here makes it seem like you are defining “high-dimensional problems” as those with “several thousand observations”, while I think you actually mean the setting with both a high-dimensional design space and several thousands of observations. Re-phrasing this could improve the clarity. > The overall acquisition strategy (described at the end of Section 2) is to concatenate the Thompson samples from all of the local models and choose the minimum. It therefore seems like this algorithm might be described as doing (“regular”) Thompson sampling over some type of approximate Bayesian model (e.g. some sort of piecewise model defined on independent and adaptively growing trust regions). Have the authors considered whether their procedure can be viewed as “regular BO”, i.e. standard Thompson sampling in a sophisticated model? ---------- Update after author response ---------- Thank you for including a high dimensional (200D) result in the author response. This seems to show good performance of TuRBO in the large-iteration regime. I have therefore bumped my score up to a 7.

[Author Response · NeurIPS 2019]

We thank the reviewers for the positive and insightful feedback. All comments will be addressed in the manuscript.

**R1+ R2** *EI vs TS, why large batches?*: The computational cost of TS scales only *linearly* with the batch size, contrasting batch EI. Moreover, TS maintains effectiveness even for large batches (see Sect. 3.7). Large batches are important for large sample budgets and, in our experience, TS scales better than batch EI. Note that TS is natural for TuRBO since we need to solve a bandit problem over the TRs. We will emphasize these points.

**R1+ R2** *Previous work on local BO*: Thank you for the pointers to a variety of interesting papers! We will revise the section on related work accordingly. We want to point out that the paper on Global Optimization with Sparse and Local Gaussian Process Models is particularly relevant, but will struggle to compete with TuRBO in high-dimensional spaces. We will mention BADS in the related work section as they also consider large evaluation budgets. However, note that several of the related papers do not consider the large-scale high-dimensional setting.

**R1** *Distribution [of optimum] known and tractable?*: This distribution is not tractable, but we can sample efficiently.

**R1** *Which of TR and TS are (most) responsible for improvement over the state of the art?*: In §3 we contrast TuRBO-1 with GP+TS that uses a global GP model. That TuRBO-1 consistently outperforms GP+TS indicates that the local modeling is most responsible for the large improvement.

**R1** *Why not compare to Regis and Shoemaker?*: Our comprehensive selection of baseline methods includes COBYLA, which is arguably one of the best well-known derivative-free TR methods.

**R1** *What does "outputscales" mean?* We meant "signal variance" and have revised the manuscript.

**R1** *Definition and evolution of TR, curse of dimensionality:* Note that TuRBO does not attempt to fill the space with smaller hypercubes. Instead, it runs several local searches simultaneously and allocates samples between them in a principled way. The domain is scaled to $[0, 1]^d$, so an initial TR with side length 1 covers the whole space.

**R1** *Lmin = $(1/2)^6$: [what is] the underlying rationale?*: Halving the size of the trust region is standard, see, e.g., Andréasson et al. An introduction to continuous optimization. The choice $(1/2)^6$ corresponds to a side length that is slightly larger than 1% of the original length.

**R2** *Why compare to COBYLA instead of BOBYQA?*: Thank you for the suggestion! We ran BOBYQA (using `nlopt`) on robot pushing (mean 9.22) and rover (mean 1.40) and it indeed performs better than COBYLA. Thus, we will replace COBYLA by BOBYQA in all experiments. Note that TuRBO still outperforms both by a large margin.

**R2** *The results of EBO for the robot experiments are quite different from the results on the original paper*: The results are consistent and we used the code from the authors. The plots in the EBO paper show standard deviation instead of standard error and median instead of mean. They also use a larger number of initial random points.

**R2** *Previous work on non-stationarity in Bayesian optimization*: Thank you for the references, we will refer to them in the revision. Note that these works do not consider our large-scale high-dimensional setting.

**R2** *It would be interesting to see the results of a standard GP+EI, maybe limiting the results to few hundred evals*: We ran GP+EI with botorch for 500 evals: the final performance is (mean 5.4) for robot pushing and (mean -6.0) on rover. This is competitive with, e.g., COBYLA, but far from the optimum. We will report the results for GP+EI to the text.

**R2** *TuRBO gets a different number of initial samples*: The plots show the performance as a function of the total number of evaluations, which includes the initial data. Note that TuRBO needs initial points for each TR to build each local GP model. We tested a different number of initial point for the baselines to maximize their performance.

**R2** *The bandit equation is purely based on the function sample and not the information/uncertainty on that region*: TS trades off exploration and exploitation effectively. To see this, note that the "function sample" can vary drastically in unexplored areas and thus will take its optimum there with a good probability.

**R3** *Benchmarks are not high-dimensional enough*: We ran TuRBO-1 and a few baselines on the 200-dimensional Ackley function (10 runs, batch 100, domain $[-5, 10]^{200}$) to illustrate that TuRBO performs well for high-dimensional problems and will add this result to the manuscript. Apart from rover, we were not aware of any non-synthetic problems in the related work with more than 50 dimensions. The 70-dimensional robot tuning problem in "Local Bayesian Optimization of Motor Skills", pointed out by R2, is worth exploring in future work.

**R3** *Can [TuRBO] be viewed as regular BO?*: We agree that this is a very interesting question for future work.

[Meta-Review · NeurIPS 2019]

The reviewers liked the paper, its results and the analysis. Please address the reviewer comments in the final version, in particular the relation to related work.